# Re-evaluating $^{14}$C dating accuracy in deep-sea sediment archives.

Bryan C. Lougheed[1], Philippa Ascough[2], Andrew M. Dolman[3], Ludvig Löwemark[4], Brett Metcalfe[5,6]

1. Department of Earth Sciences, Uppsala University, Sweden.

2. Scottish Universities Environmental Research Centre, Glasgow, Scotland, UK.

3. Alfred Wegener Institute, Helmholtz Centre for Polar and Marine Research, Potsdam, Germany

4. Department of Geosciences, National Taiwan University, Taipei, Taiwan.

5. Department of Earth Sciences, Vrije Universiteit Amsterdam, the Netherlands.

6. LSCE-IPSL, CEA-CNRS-UVSQ, Université Paris-Saclay, Gif-sur-Yvette, France

Corresponding author: B.C. Lougheed (bryan.lougheed@geo.uu.se)

## Abstract

The current geochronological state-of-the-art for applying the radiocarbon ($^{14}$C) method to deep-sea sediment archives lacks key information on sediment bioturbation. Here, we apply a sediment accumulation model that simulates the sedimentation and bioturbation of millions of foraminifera, whereby realistic $^{14}$C activities (i.e. from a $^{14}$C calibration curve) are assigned to each single foraminifera based on its simulation timestep. We find that the normal distribution of $^{14}$C age typically used to represent discrete-depth sediment intervals (based on the reported laboratory $^{14}$C age and measurement error) is unlikely to be a faithful reflection of the actual $^{14}$C age distribution for a specific depth interval. We also find that this deviation from the actual $^{14}$C age distribution is greatly amplified during the calibration process. Specifically, we find a systematic underestimation of total geochronological error in many cases (by up to thousands of years), as well as the generation of age-depth artefacts in downcore calibrated median age. Even in the case of "perfect" simulated sediment archive scenarios, whereby sediment accumulation rate (SAR), bioturbation depth, reservoir age and species abundance are all kept constant, the $^{14}$C dating and calibration process generates temporally dynamic median age-depth artefacts, on the order of hundreds of years – whereby high SAR scenarios (40 cm kyr$^{-1}$ and 60 cm kyr$^{-1}$) are not immune. Such age-depth artefacts can be especially pronounced during periods corresponding to dynamic changes in the Earth's $\Delta^{14}$C, when single foraminifera of varying $^{14}$C activity can be incorporated into single discrete-depth sediment intervals. For certain lower SAR scenarios, we find that downcore discrete-depth true median age can systematically fall outside calibrated age range predicted by the $^{14}$C dating and calibration process, thus leading to systematically inaccurate age estimations. In short, our findings suggest the possibility of $^{14}$C-derived age-depth artefacts in the literature. Furthermore, since such age-depth artefacts are likely to coincide with large-scale changes in global $\Delta^{14}$C, which themselves can coincide with large-scale changes in global climate (such as the last deglaciation), $^{14}$C-derived age-depth artefacts may have been

 previously incorrectly attributed to changes in SAR coinciding with global climate. Our study highlights the need for the development of improved deep-sea sediment $^{14}$C calibration techniques that include an *a priori* representation of bioturbation for multi-specimen samples.

## 1.0 Introduction

### 1.1 Background and rationale

For over half a century, radiocarbon ($^{14}$C) dating has been applied to deep sea sediment archives. The material that is typically analysed from these archives consists of the calcareous tests of foraminifera. The minimum amount of material required for viable $^{14}$C analysis has meant that researchers have had to pick tens to hundreds of individual foraminifera specimens (depending on specimen size) from a single discrete-depth core interval (typically 1 cm of core depth) and combine these into a single sample for analysis. Such multi-specimen samples are likely to be heterogeneous in $^{14}$C activity (i.e. combine individual specimens of varying true age). The $^{14}$C laboratory measurement (and reported machine error) applied to such an amalgamated multi-specimen sample will simply represent the mean $^{14}$C activity of the total carbon of all individual specimens. Consequently, the true intra-sample $^{14}$C age heterogeneity of a sample is concealed from the researcher. Failure to consider the actual $^{14}$C age heterogeneity of multi-specimen samples can lead to downcore $^{14}$C age artefacts when post-depositional processes mix foraminifera with differing $^{14}$C activities, especially during periods coinciding with periods of dynamic $\Delta^{14}$C history of the Earth. Furthermore, one must also take into consideration that younger specimens within a sample contribute exponentially more to the sample's mean $^{14}$C activity than older specimens do, a process referred to as the isotope mass balance effect (Erlenkeuser, 1980; Keigwin and Guilderson, 2009), due to $^{14}$C being a radioactive isotope (specimen $^{14}$C activity decreases exponentially with the passing of time).

Systematic bioturbation has long been recognised as an inherent feature of deep-sea sediment archives (Bramlette and Bradley, 1942; Arrhenius, 1961; Olausson, 1961). Long-established mathematical models of bioturbation in deep-sea sediment archives consider the uppermost ~10 cm of a sediment archive to be uniformly mixed due to active bioturbation - the bioturbation depth (BD) (Berger and Heath, 1968; Berger and Johnson, 1978; Berger and Killingley, 1982). The presence of such a BD has been supported by the detection of a uniform mean age in the uppermost intervals of sediment archives (Peng et al., 1979; Trauth et al., 1997; Boudreau, 1998; Teal et al., 2008) and by the $^{14}$C analysis of single foraminifera (Lougheed et al., 2018). The total range of single specimen ages mixed within the BD is dependent upon two main factors: the depth of the BD itself, and the sediment accumulation rate (SAR), both of which can exhibit spatiotemporal variation due to environmental and biological factors (Müller and Suess, 1979; Trauth et al., 1997). The presence of uniform mixing within the BD throughout the sedimentation history of a deep-sea sediment archive ultimately results, in the case of temporally constant SAR and BD, in the single specimen population of discrete

sediment intervals being characterised by an exponential probability density function (PDF) for true age, with a maximum probability for younger ages and a long tail towards older ages. The existence of such a distribution has been supported by the post-depositional mixing of tephra layers (Bramlette and Bradley, 1942; Nayudu, 1964; Ruddiman and Glover, 1972; Abbott et al., 2018) and the smoothing out of the downcore mean signal (Guinasso and Schink, 1975; Pisias, 1983; Schiffelbein,

1984; Bard et al., 1987; Löwemark et al., 2008; Trauth, 2013), the smoothing of which can change downcore in tandem with foraminiferal abundance changes (Ruddiman et al., 1980; Peng and Broecker, 1984; Paull et al., 1991; Löwemark et al., 2008). If SAR, BD and the $\Delta^{14}$C history of the planet were all to be temporally constant, then the idealised $^{14}$C activity PDF of each discrete depth (expressed as, e.g., the $^{14}$C/$^{12}$C ratio or normalised as fraction modern [F$^{14}$C]) would, therefore, exhibit

the combination of two exponential functions (the exponential PDF of true age plus the exponential PDF of $^{14}$C activity vs time predicted by the half-life of $^{14}$C). However, the distribution of the $^{14}$C activity PDF is made complicated by the fact that $^{14}$C activity vs time is not always the exact exponential function that would be predicted by the radioactive half-life of $^{14}$C, seeing as the Earth's carbon reservoir exhibits a dynamic $\Delta^{14}$C history, as demonstrated by temporal changes in

atmospheric $^{14}$C activity (Suess, 1955, 1965; de Vries, 1958; Reimer et al., 2013). These changes are brought about by changes in $^{14}$C production in the atmosphere in combination with climatic and oceanic influence upon the carbon cycle (Craig, 1957; Damon et al., 1978; Siegenthaler et al., 1980). Furthermore, non-uniform mixing of the oceans can contribute to temporal changes in local water $^{14}$C activity at a given coring site, further affecting the idealised PDF shape.

When applying the $^{14}$C method to sediment core material, researchers represent the $^{14}$C activity of a discrete-depth interval using a normal (Gaussian) distribution, based on the conventional mean $^{14}$C age (a reporting convention for $^{14}$C activity) and measurement error reported by the $^{14}$C laboratory (Stuiver and Polach, 1977). In some cases, this $^{14}$C age normal distribution is widened by researchers to also incorporate a reservoir age uncertainty, but it remains a normal distribution. This normal

distribution of $^{14}$C age is subsequently calibrated using a suitable reference record of past $\Delta^{14}$C (e.g. those produced by the *IntCal* group), allowing researchers to arrive at an estimation of the discrete depth interval's true (i.e. calendar) age. Such an approach inherently excludes the effects of bioturbation, because one would not expect a normal $^{14}$C age distribution to be representative of a discrete depth interval, for the reasons described in the previous paragraph. Currently, systematic

investigation is lacking into whether neglecting to include the effects of bioturbation has significant impact upon the interpretative accuracy of $^{14}$C dating as it is currently applied in palaeoceanography, i.e. if it may ultimately lead to spurious geochronological interpretations.

**1.2 Experimental design**

Here, we we take advantage of computer modelling to construct an ideal experimental design whereby we can evaluate how the current $^{14}$C state-of-the-art within palaeoceanography would work in the case of best-case sediment conditions. Such best-case conditions do not exist in in the field, meaning that a computer modelling environment can uniquely be used to create such a best-case scenario, which is ideal for testing the current state-of-the-art. We use the $\Delta^{14}$C-enabled, single-specimen SEdiment AccuMUlation Simulator (SEAMUS) (Lougheed, 2020). This model uses the long-established understanding of bioturbation as included in existing bioturbation models (Trauth, 2013; Dolman and Laepple, 2018), but differs in that it explicitly simulates the accumulation and bioturbation of single foraminifera, each with individually assigned $^{14}$C activities, to create a synthetic sediment archive history. Subsequently, current palaeoceanographic subsampling and $^{14}$C dating practices are virtually applied to the 1 cm discrete-depth sediment intervals of the model's outputted synthetic archive, resulting in discrete-depth $^{14}$C ages and calibrated ages that are representative of the existing palaeoceanographic state-of-the-art. These results are subsequently compared to the actual discrete-depth true age distributions within the model, allowing us to quantitatively evaluate contemporary palaeoceanographic $^{14}$C dating and calibration techniques. By keeping multiple model input parameters constant, we can construct an experimental environment whereby we have full control over the degrees of freedom. This modelling approach allows us to test, at a most fundamental level, the accuracy of the current $^{14}$C dating state-of-the-art as applied to deep-sea sediments.

## 2.0 Method

### 2.1 The synthetic core simulation

The SEAMUS model (Lougheed, 2020) synthesises $n$ number of single foraminifera raining down from the water column per simulation timestep, whereby $n$ is the capacity of the synthetic sediment archive being simulated (analogous to sediment core radius) scaled to the SAR of the timestep as predicted by an inputted age-depth relationship (Lougheed, 2020). To provide good statistics, all simulations use a timestep of 5 years and $10^4$ synthetic foraminifera per cm core depth. An abundance of $10^4$ specimens per cm is also similar to a best-case scenario value for a particular sample in the field (Broecker et al., 1992).

In each timestep, all newly created single foraminifera are assigned an age (corresponding to the timestep), a sediment depth (according to the age-depth input), as well as a $^{14}$C age (in $^{14}$C yr BP) and normalised $^{14}$C activity (in F$^{14}$C) based on *Marine13* (Reimer et al., 2013) after the application of a prescribed reservoir age for the timestep. For older sections of the *Marine13* calibration curve, where only 10 year timesteps are available, linear interpolation is used to provide a 5 year $^{14}$C activity timestep resolution. Within SEAMUS, all single foraminifera older than the oldest available age within the chosen calibration curve (in this case *Marine13*) are assigned the same $^{14}$C activity: that of the analytical blank, which must be set in the simulation. In this way, the model incorporates the

principles of [14]C dating, whereby individual very old foraminifera contained within a sample will contribute a [14]C signal equivalent to the analytical blank. Here, we choose to set the the simulation's analytical blank value to 46806 [14]C yr BP (more precisely the $F^{14}C$ equivalent thereof), which corresponds to the lowest activity level in the *Marine13* calibration curve. The analytical blank activity in most laboratories is somewhat lower (e.g., >50000 [14]C yr BP), but we have no way of accurately applying an activity to single foraminifera older than the oldest value contained within *Marine13*. Rather than infer a $\Delta^{14}C$ history beyond the limit of *Marine13*, we simply set the analytical blank in our simulation to 46806 [14]C yr BP. In some scenarios we wish to investigate parameters within an experimental construct with temporally constant $\Delta^{14}C$, and in such scenarios we assign [14]C activity (as $F^{14}C$) as follows: $F^{14}C(t) = e^{(\,[t+R(t)]\,/\,-8267\,)}$, where $t$ is the single foraminifera age in years before 1950 CE, and $R(t)$ is the reservoir age for age $t$.

After the creation of all new single foraminifera within the synthetic core for a specific timestep, bioturbation is simulated. Specifically, for each timestep the depth values corresponding to all simulated foraminifera within the contemporaneous BD are each assigned a new depth by way of uniform random sampling of the BD interval. In this way, uniform mixing of foraminifera within the BD is simulated following established understanding of bioturbation (Berger and Heath, 1968; Trauth, 2013). All of the aforementioned processes are repeated for every simulation timestep until such point that the end of the age-depth input (i.e. the final core top) is reached. All simulations are initiated at 70 ka (in true age) in order to confidently exclude the influence of model spin-up effects upon our period of interest (0 – 45 ka), given the possibility of a given cm of sediment to have a long-tail of older foraminifera specimens. While SEAMUS can in principle be run on a local machine, to save time multiple simulations were run in parallel on a computing cluster provided by the Swedish National Infrastructure for Computing (SNIC) at the Uppsala Multidisciplinary Centre for Advanced Computational Science (UPPMAX).

**2.2 Virtual discrete-depth analysis**

After the completion of the synthetic core simulation, synthetic foraminifera (and corresponding values for true age, $F^{14}C$, and [14]C age) are picked from each discrete 1 cm interval of the sediment core. In this study, we assume best-case scenarios where it is possible to pick all whole foraminifera contained within the sediment intervals. Subsequently, each of these picked 1 cm samples also undergoes a synthetic [14]C determination analogous to a perfect accelerator mass spectrometry (AMS) measurement, whereby it is assumed that the AMS determination perfectly reproduces the mean [14]C activity (in $F^{14}C$) of the sample. Within the discrete-depth subsampling simulation, this mean [14]C activity is calculated by taking the mean of all $F^{14}C$ values of all the single foraminifera contained within the picked sample. As mentioned in Section 2.1, the analytical blank is already included when

assigning ¹⁴C to single foraminifera, meaning that the influence of the analytical blank upon sample AMS measurements is incorporated.

Using the Libby half-life, a sample's mean F¹⁴C value is also reported as a conventional ¹⁴C age determination (in ¹⁴C yr). All such synthetic determinations are assigned a synthetic 1σ measurement error analogous to a best-case scenario laboratory counting error for a large sample. The prescribed synthetic measurement error ranges from 30 ¹⁴C yr in the case of near-modern samples to 500 ¹⁴C yr in the case of samples nearing the blank value. Specifically, when assigning measurement errors to synthetic AMS determinations, a ¹⁴C determination of 1.0 F¹⁴C is assumed to have a measurement error of 30 ¹⁴C yr, and a determination with the F¹⁴C value $e^{(blankvalue-1)/-8033}$ (i.e. one ¹⁴C yr younger than the blank value) is assumed to have a measurement error of 500 ¹⁴C yr. Errors (in ¹⁴C yr) for intermediate dates are linearly interpolated to F¹⁴C.

The synthetic laboratory ¹⁴C determinations and associated measurement uncertainties for each 1 cm discrete-depth sample are subsequently converted to calibrated years within SEAMUS using the embedded MatCal (v 2.6) ¹⁴C calibration software (Lougheed and Obrochta, 2016), the *Marine13* calibration curve (Reimer et al., 2013) and a prescribed reservoir age (according to the scenario – see following sections), to produce a calibrated age probability density function (PDF) and 95.4% highest posterior density (HPD) credible interval(s) for every cm core depth, i.e. analogous to what would be typically produced using contemporary palaeoceanography methods in the case of every discrete cm of core depth being exhaustively ¹⁴C dated. The MatCal software calibrates ages in F¹⁴C space, resulting in an accurate calibration, especially in the case of older samples or samples with large uncertainty.

**3.0 Best-case scenario simulations**

In order to investigate the baseline accuracy when applying ¹⁴C dating to deep-sea sediment cores, the first simulations in this study consider a number of best-case scenarios. Essentially, we seek to test how well the current application of ¹⁴C within palaeoceanography would function in the case of such a best-case scenario, thus testing the current state-of-the-art at a most fundamental level. In such simulations, we assume that *Marine13* constitutes a perfect reconstruction of past surface-water ¹⁴C activity at the synthetic core site, and we therefore employ a temporally constant reservoir age (ΔR = 0 ¹⁴C yr). Furthermore, we assume a scenario involving synthetic sediment cores with temporally constant SAR and BD, and we also assume that the synthetic core is made up of a single planktonic foraminiferal species with a temporally constant abundance (10⁴ cm⁻¹) and specimen size. A total of five best case scenarios are carried out, with five different SAR scenarios (5, 10, 20, 40 and 60 cm kyr⁻¹). The BD is set to 10 cm in all cases, following established understanding of global BD (Trauth et al., 1997; Boudreau, 1998). In this scenario, we also assume perfection in sub-sampling, i.e. the possibility to exhaustively sample all foraminifera material from each 1 cm discrete-depth interval

when picking for multi-specimen samples, thus excluding noise due to small sample sizes. The results of these five scenarios are visualised in Fig. 1 and Fig. S1-S5.

A second set of best-case scenarios takes into account that relatively older foraminifera contained within a given discrete depth of core sediment will have accumulated a longer residence time in the active bioturbation depth. Due to their longer residence time in the active bioturbation depth, these foraminifera are more likely to be broken and/or partially dissolved (Rubin and Suess, 1955; Ericson et al., 1956; Emiliani and Milliman, 1966; Barker et al., 2007), and are thus less likely to be picked by

palaeoceanographers who preferentially pick whole/unbroken foraminifera specimens for analysis. In this way, palaeoceanographers exclude the oldest, least-well preserved fraction of the sediment. An indication of the BD residence time of single specimens for a given 1 cm discrete depth is shown in Fig. 2 for all five simulated SAR scenarios, along with the median and 90[th] percentile residence time. The percentage of broken specimens within the sediment archive is chiefly governed by the

aforementioned BD residence time, bottom water chemistry (Bramlette, 1961; Berger, 1970; Parker and Berger, 1971), and the susceptibility of a particular foraminifera species to dissolution/breakage (Ruddiman and Heezen, 1967; Boltovskoy, 1991; Boltovskoy and Totah, 1992). Previous studies have indicated that the percentage of foraminifera exhibiting test breakage for typically analysed species at locations above the lysocline can hover around 10% (Le and Shackleton, 1992). In the second set of

best-case scenarios we, therefore, exclude from the picking process for each 1 cm discrete depth all foraminifera with a number of bioturbation cycles greater than the 90[th] percentile for that particular discrete depth. This broken foraminifera percentage of 10% is applied to all five SAR scenarios (5, 10, 20, 40, 60 cm kyr[-1]) in a second set of best case scenarios, shown in Fig. 3 and Fig. S6-S10. One should be aware, however, that BD residence time likely varies with SAR itself: when sediment

accumulation is slower, single specimens remain in the BD for relatively longer than in the case of faster SAR (Bramlette, 1961).

### 3.1 $^{14}$C age artefacts

Radiocarbon analysis focuses on determining the mean $^{14}$C activity of a particular sample, which is reported together with an associated analytical error. This mean activity of samples is often

considered in the literature as conventional $^{14}$C age in $^{14}$C yr BP. Conventional $^{14}$C age, a unit of convenience, is linear vs time, whereas $^{14}$C activity is actually exponential vs time, due to $^{14}$C being a radioactive isotope. Therefore, with increasing age heterogeneity of a sample, we can expect increased offset between the AMS conventional $^{14}$C age of a sample (the mean measured activity of the homogenised sample reported as conventional age) and the idealised mean of the conventional $^{14}$C

ages of all single foraminifera within the sample. In Fig. 1, we compare the simulated AMS mean conventional $^{14}$C age calculated for each discrete depth to the idealised mean $^{14}$C age (based on the mean value of all single foraminifera conventional $^{14}$C ages contained within a sample). The resulting

offset can help shed light upon how the measurement of age-heteregenous material is inherently biased towards younger (higher $^{14}$C activity) specimens contained within the sample. We find that the AMS mean $^{14}$C age is generally younger than the idealised mean $^{14}$C age in all cases. This effect can be attributed to the fact that younger foraminifera within a heterogeneous sample contribute exponentially more to a sample's mean $^{14}$C activity (what the measurement process is actually analysing) than older foraminifera do. This bias towards younger foraminifera is most apparent in cases with large intra-sample heterogeneity, such as in scenarios with lower SAR (Fig. 1a), and is also reduced somewhat in the case of more broken foraminifera (Fig. 3a), due to lesser older foraminifera being picked, thus reducing the age heterogeneity. In the case of the highest SAR scenarios (> 40 cm kyr$^{-1}$) the aforementioned bias is insignificant in a practical sense, in that it falls within the typical $^{14}$C measurement error. For all scenarios, superimposed upon the general bias are artefacts of the Earth's dynamic $\Delta^{14}$C history, caused by foraminifera from times of markedly differing $\Delta^{14}$C to be mixed together into a single sample, thus altering a sample's $^{14}$C activity distribution and causing downcore dynamic offsets between AMS mean $^{14}$C age and idealised mean $^{14}$C age. The most pronounced example of these artefacts can be seen during known periods of dynamic $\Delta^{14}$C, such as during the Laschamps geomagnetic event (ca. 40~41 ka) (Guillou et al., 2004; Laj et al., 2014), when a large spike in atmospheric $^{14}$C production occurred (Muscheler et al., 2014). We note that our simulations assign single foraminifera $^{14}$C activity using the *Marine13* calibration curve, while newer records of $\Delta^{14}$C (Cheng et al., 2018) suggest that the Laschamps $\Delta^{14}$C excursion may have been of greater magnitude than was previously thought. A larger excursion would generate even more pronounced $^{14}$C artefacts in the downcore, multi-specimen, discrete-depth record. Furthermore, there may exist as yet undiscovered, past short-lived excursions in $\Delta^{14}$C (Miyake et al., 2012, 2017; Mekhaldi et al., 2015).

We can also visualise how well a sample's $^{14}$C activity probability distribution function (PDF) is represented by a distribution based on its mean AMS-measured $^{14}$C activity and 1σ measurement error. This visualisation is shown on the vertical axes of Fig. 1d-i and Fig. 2d-i for a number of simulated discrete depths for the different SAR scenarios with a BD of 10 cm. It can be clearly seen that that the normal distribution derived from a sample's AMS mean measurement and associated uncertainty is a poor representation of a sample's actual $^{14}$C activity distribution.

## 3.2 Calibration amplifies $^{14}$C age distribution mischaracterisation

When estimating a true age distribution for a particular sample, researchers calibrate a normal distribution of $^{14}$C age using suitable calibration curve (in this case *Marine13*). As discussed in the previous section, the aforementioned normal distribution of $^{14}$C activity derived from the measurement mean and machine error is not a faithful representation of the actual $^{14}$C activity distribution for a particular discrete depth. Such a misrepresentation has the potential to be further amplified during the calibration process itself, potentially resulting in a poor estimation of a discrete

depth's 95.4% age range and/or median age, the latter of which is often used to calculate e.g. sedimentation rates, or represents the region of highest probability which will steer age-depth modelling routines. In Fig. 1b (0% broken foraminifera) and Fig. 3b (10% broken foraminifera), we show the offset between each discrete depth's true median age, and the corresponding median age derived from [14]C calibration process. We find large offsets for all constant SAR scenarios, ranging from ~200 years in the case of the the 60 cm kyr[-1] scenario, to up to ~700 years in the case of the 5 cm kyr[-1] scenario. In certain low SAR scenarios that coincide intervals of the calibration curve that are highly resolved (e.g. the late Holocene), the discrete-depth true median age can consistently fall outside the 68.2% age range predicted by the [14]C dating and calibration process. A 68.2% certainty suggests that, statistically, the true median will fall outside of the 68.2% calibrated age range in only 31.8% of cases, but in the case of the 5 cm kyr[-1] scenario (Fig. S1), the true median falls outside of the 68.2% calibrated age range for 84% of the discrete depths spanning the 5 to 0 cal ka period. In the case of 10% broken foraminifera, this effect is reduced.

All offsets for all scenarios vary dynamically downcore, meaning that they can potentially cause spurious interpretations of changes in SAR. Furthermore, as these offsets occur during periods of dynamic $\Delta^{14}C$, which can be caused by large-scale changes in the carbon cycle caused by climate shifts (such as during the last deglaciation), it is possible that some apparent changes in SAR in the palaeoceangraphic literature may have been erroneously attributed to climate processes, when they may be (partially) an artefact of the current application of [14]C dating and calibration within palaeoceanography.

Using the simulation output, it is also possible to quantitatively estimate how well the current [14]C dating and calibration state-of-the-art applied within palaeoceanography estimates the true age range contained within discrete-depth sediment intervals. The offset between the calibrated 95.4% age range and the true 95.4% age range for each discrete depth for all SAR scenarios is shown in Fig. 1c (0% broken foraminifera) and Fig. 3c (10% broken foraminifera) and is further visualised for all scenarios in Fig. S1-S10. For the lower SAR scenarios, the current application of [14]C dating within palaeoceanography significantly underestimates the total age range contained within each discrete-depth, by many thousands of years. The underestimation is less in the case of the scenario with 10% broken foraminifera. In the case of higher SAR scenarios, the discrete-depth 95.4% age range predicted by the [14]C calibration process is similar to that of the discrete depth 95.4% age range of the sediment itself. In some cases with very high SAR, the [14]C calibration process actually overestimates the 95.4% age range (e.g. Fig. 1e, Fig. 3e, Fig. S5 and Fig. S10).

**3.3 The influence of the analytical blank**

A general consequence of bioturbation and the subsequent mixing of single foraminifera specimens is that older foraminifera become systematically mixed upwards throughout the sedimentation history of

a sediment archive. This general mixing can have a particular consequence near the analytical limit of the $^{14}$C method, in that foraminifera with a $^{14}$C activity that is lower than a laboratory's analytical sensitivity can become mixed into samples. $^{14}$C determinations with a $^{14}$C age that is older than the established $^{14}$C blank value (i.e. they fall below the detection limit of the analytical process) are commonly referred to as "$^{14}$C-dead". Within older intervals of heterogeneous deep-sea sediment archives, it is possible that a sample with an apparent measured $^{14}$C age younger than the $^{14}$C blank value can already contain a significant proportion of $^{14}$C-dead foraminifera. The presence of these $^{14}$C-dead specimens within a sample will bias the sample's apparent measured $^{14}$C age towards a too young value, because they will contribute a $^{14}$C activity to the sample that is equivalent to the laboratory's analytical blank. Such artefactually young $^{14}$C ages could ultimately erroneously be interpreted as age-depth features. In Table 1, the very first downcore occurrence of at least one simulated $^{14}$C-dead foraminifer is detailed for each of the aforementioned constant SAR scenarios introduced in Section 3.0. In the case of low SAR scenarios with 0% broken foraminifera, $^{14}$C-dead foraminifera are already present in discrete-depth samples with apparent AMS ages that would normally be considered well above the $^{14}$C blank value, e.g. an apparent AMS age of 22647 $^{14}$C yr BP in the case of 5 cm kyr$^{-1}$, and 33747 $^{14}$C yr BP in the case of 10 cm kyr$^{-1}$. However, the contribution of $^{14}$C-dead foraminifera at these levels may still be insignificant. The exact percentage contribution of $^{14}$C-dead foraminifera to discrete depth AMS determinations is, therefore, detailed in Fig. 4a, 4c, 4e, 4g and 4i. From this analysis, it transpires that the first occurrence of at least 1% contribution of $^{14}$C-dead foraminifera to discrete-depth AMS determinations occurs in the case of AMS ages of 39158 $^{14}$C yr BP and 43601 $^{14}$C yr BP, respectively for the 5 cm kyr$^{-1}$ and 10 cm kyr$^{-1}$ scenarios. The percentage increases quickly further downcore. In the case of scenarios involving 10% broken foraminifera, older foraminifera within discrete-depth sediment intervals are no longer whole, and therefore not picked for samples by a palaeoceanographer preferring whole specimens. The consequence of this effect is that the first occurrence of picked $^{14}$C-dead whole foraminifera occurs much further downcore (Table 1, Fig. 4b, 4d, 4f, 4h and 4j). This finding further underlines the importance of understanding foraminifera preservation conditions for particular species and/or water chemistry, and the associated consequences for $^{14}$C dating.

As motivated in the method section, for practical reasons we have set the $^{14}$C analytical blank value at 46806 $^{14}$C yr BP within our model simulations. The laboratory blank value in most laboratories is around ~50000 $^{14}$C yr BP, or even greater, depending on sample size, preparation conditions and measurement capability. For such greater blank values, essentially the same curves as shown in Fig. 4 would apply (i.e. assuming there are no as of yet undiscovered, large $\Delta^{14}$C excursions around the period of the blank age), but shifted further to the right on the x-axis. In other words, researchers interested in interpreting Fig. 4 in the case of an analytical blank of 50000 $^{14}$C yr BP should simply

shift the curves to the right such that the 100% $^{14}$C-dead contribution exactly coincides with 50000 $^{14}$C yr BP on the x-axis.

**4.0 Dynamic sediment core scenarios**

The multiple sediment archive scenarios carried out in Section 3.0 all involved best-case input parameters with constant SAR. In Fig. 5, we carry out four scenarios to investigate the influence of stepwise changes in the following four input parameters: (1) SAR; (2) BD; (3) species abundance and; (4) reservoir age ($\Delta$R). In each of the four scenarios, one of the aforementioned input parameters is varied at a certain time while the other three are kept constant (Figs. 5a-d). In this way, the influence of one of the dynamic input parameters can be independently judged. To further ensure the ability to independently judge the dynamic sediment input parameters, in these scenarios we do not employ a dynamic $\Delta^{14}$C history using *Marine13,* but instead assign $^{14}$C activities to foraminifera using a constant $\Delta^{14}$C history (with an added constant 400 yr reservoir age). This constant $\Delta^{14}$C history is assigned as detailed in the method (Section 2.1). For the calibration process, we also constructed a calibration curve with same aforementioned constant $\Delta^{14}$C (also with an added constant 400 yr reservoir age), whereby the confidence interval sizes of *Marine13* are copied for incorporating a realistic calibration uncertainty. The scenario with dynamic $\Delta$R (Fig. 5d) is simulated upon the foraminifera by additionally subtracting ($\Delta$R = -100) or subtracting ($\Delta$R = +100) to/from the $^{14}$C age of simulated foraminifera respectively younger or older than 20 ka. During the simulated picking and calibration processes, it is assumed that the researcher is aware of the change in $\Delta$R and, during calibration, applies a $\Delta$R of -100 to all discrete depths shallower than 204 cm, and a $\Delta$R of +100 to all discrete depths deeper than 204 cm.

The simulations using dynamic parameter inputs demonstrate that temporal changes in any of the four main input parameters (SAR, BD, species abundance, $\Delta$R) can result in the generation of $^{14}$C-induced age-depth artefacts in the discrete-depth domain, due to the median calibrated age dynamically deviating from the true median age downcore (Fig. 5f). We also note that the changes in the input parameters can cause the $^{14}$C dating and calibration process to generate artefacts in the over- or underestimation of the true 95.4% age range of the sample by the calibration process, artefacts which are superimposed upon a long-term change in the underestimation of the true age range of the sample caused by a long term change in the confidence intervals in the calibration curve (Fig 5g). Specifically regarding $\Delta$R, the current method for correcting for reservoir age during calibration, which we apply in this simulation, involves subtracting the $\Delta$R from the the AMS date just prior to calibration. This method poses a particular challenge for periods near temporal changes in $\Delta$R, where multi-specimen samples will incorporate single foraminifera with varying individual $\Delta$R values. The blanket application of a single $\Delta$R correction to the entire sample fails to adequately represent the $\Delta$R heterogeneity of the foraminifera population.

The influence of the various dynamic parameters upon the $^{14}$C dating and calibration process, as outlined in Fig. 5, represent further sources of age-depth bias in addition to the large biases caused by dynamic $\Delta^{14}$C history previously outlined in Section 3.0. Furthermore, as has been detailed in previous studies, changes in abundance and bioturbation depth can in themselves also cause additional general age-depth artefacts, no matter what geochronological method is being used (independent of the $^{14}$C method), (Bard, 2001; Löwemark and Grootes, 2004; Löwemark et al., 2008; Lougheed, 2020). Such effects can be seen in age-depth artefacts also visible in the true median age for the dynamic BD scenario (Fig. S12) and the dynamic abundance scenario (Fig. S13). Such artefacts occur in addition to the artefacts related to the $^{14}$C dating and calibration process outlined in this study.

Researchers should be aware that periods of long-term climate change can cause many input parameters to change in concert. For example, the last deglaciation in the North Atlantic is known to be characterised by highly dynamic $\Delta^{14}$C (Stuiver et al., 1986; Reimer et al., 2013), dynamic reservoir age (Austin et al., 1995; Waelbroeck et al., 2001; Butzin et al., 2020) and dynamic foraminiferal abundance (Ruddiman and McIntyre, 1981). It is possible that all of these parameters can combine at once to produce very large age-depth artefacts, which could lead to spurious interpretations regarding the relationship between, e.g., the last deglaciation and the perceived magnitude of associated SAR change.

**5.0 Conclusion**

This study demonstrates the possibility for the current $^{14}$C dating and calibration method, as it is applied to multi-specimen samples within palaeoceanography, to produce age-depth artefacts, even in the case of best-case sediment archives where SAR, BD, species abundance and reservoir age are all constant. We also find that high SAR sediment archives (40 cm kyr$^{-1}$ and 60 cm kyr$^{-1}$) are not immune to the generation of age-depth artefacts during the $^{14}$C dating and calibration process. Additional age-depth artefacts can be generated in the case of real-world sediment archives where the aforementioned SAR, BD, species abundance and reservoir age processes are inherently dynamic. Researchers should be aware, therefore, of the possible existence of such artefacts when interpreting deep-sea sediment geochronologies developed using $^{14}$C methods applied to multi-specimen samples. Key to understanding the possible existence of such artefacts is a good quantification of the possible magnitude of temporal change in both foraminiferal abundance and preservation conditions, as well as awareness of the possibility of changes in local $^{14}$C activity due to the influence of dynamic $\Delta^{14}$C and reservoir age. It may also be necessary to revisit existing studies and re-evaluate the magnitude of changes in deep-sea sediment SAR inferred from $^{14}$C-based geochronologies, especially close to periods of dynamic $\Delta^{14}$C and/or dynamic foraminiferal abundance. These $^{14}$C-specific artefacts should be considered in addition to previously highlighted general age-depth artefacts that can occur in sedimentary records (Bard, 2001; Löwemark and Grootes, 2004; Löwemark et al., 2008; Lougheed,

2020). One should also consider that paired analysis of mulispecimen samples for both [14]C and another proxy could lead to a signal offset between the two proxies due to the [14]C method, as currently applied within palaeoceanography, being prone to the generation of age artefacts as outlined in this study.

**6.0 Outlook and future research**

We demonstrate that the failure to take into account the effect of bioturbation upon the ([14]C) age distribution of foraminifera in multi-specimen samples sourced from deep-sea archives can lead to spurious age interpretations, especially during the [14]C calibration process. We propose, therefore, that the [14]C calibration process for deep-sea sediment archives could be improved in future studies through the development of a new [14]C calibration method including bioturbation *a priori*, seeing that no information regarding bioturbation is included in the current palaeoceanographic state-of-the-art. This new approach would involve constructing a representative distribution for [14]C age that includes *a priori* information regarding the approximate SAR and BD of the sediment archive, while also taking into account some basic information regarding possible temporal changes in species abundance and $\Delta R$. Such a future development would go some way to providing more realistic uncertainties (i.e. 95.4% age range) to [14]C-derived age-depth geochronologies in deep-sea sediment archives.

Finally, we note that increased automation and cost-effectiveness in [14]C analysis of ultra-small carbonate samples (Ruff et al., 2010; Lougheed et al., 2012; Wacker et al., 2013b, 2013a) can allow for the parallel measurement of $\delta^{18}O$, $\delta^{13}O$ and [14]C on a single foraminifer of suitable size (Lougheed et al., 2018), thereby allowing for the extraction of both age and palaeoclimate data from single foraminifera in a manner that is independent of the sediment depth and bioturbation aspects of deep-sea sediment archives.

**Author contributions**

BCL carried out the model runs, with scenarios conceived with input from BM. BCL wrote the manuscript with input from the co-authors.

**Acknowledgements**

The Swedish National Infrastructure for Computing (SNIC) at the Uppsala Multidisciplinary Centre for Advanced Computational Science (UPPMAX) provided computing resources. Two anonymous referees and editor Irka Hajdas are thanked for their contribution to the online discussion forum. Their input helped to significantly improve the manuscript.

**Financial support**

This work was funded by Swedish Research Council (Vetenskapsrådet – VR) Starting Grant number 2018-04992 awarded to BCL.. BM was supported by a Laboratoire d'excellence (LabEx) of the Institut Pierre-Simon Laplace (Labex L-IPSL), funded by the French Agence Nationale de la Recherche (grant no. ANR-10-LABX-0018). AMD was supported by the German Federal Ministry of Education and Research (BMBF) as a Research for Sustainability initiative (FONA) through the PalMod project (FKZ: 01LP1509C). LL acknowledges support from Ministry of Science and Technology (06-2116-M-002-021 to LL), and the Featured Areas Research Center Program within the framework of the Higher Education Sprout Project by the Ministry of Education (MOE) of Taiwan.

**Review statement**

This paper was edited by Irka Hajdas and reviewed by two anonymous referees.

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

| | First downcore occurrence of "$^{14}$C-dead" foraminifera | | | | | | | |
|---|---|---|---|---|---|---|---|---|
| | *0 % broken foraminifera scenario* | | | | *10% broken foraminifera scenario* | | | |
| | *Discrete depth (cm)* | *Median true age (yr)* | *AMS $^{14}$C age ($^{14}$C yr BP)* | *Median $^{14}$C calibrated age (cal yr BP)* | *Discrete depth (cm)* | *Median true age (yr)* | *AMS $^{14}$C age ($^{14}$C yr BP)* | *Median $^{14}$C calibrated age (cal yr BP)* |
| *SAR 5 cm kyr$^{-1}$ BD 10 cm* | 133-134 | 26110 | 22647 | 26493 | 237-238 | 46690 | 44096 | 46833 |
| *SAR 10 cm kyr$^{-1}$ BD 10 cm* | 375-376 | 37250 | 33747 | 37654 | 486-487 | 48260 | 45422 | 48396 |
| *SAR 20 cm kyr$^{-1}$ BD 10 cm* | 900-901 | 44855 | 41973 | 45002 | 986-987 | 49125 | 46090 | 49186 |
| *SAR 40 cm kyr$^{-1}$ BD 10 cm* | 1894-1895 | 47285 | 44582 | 47383 | 1987-1988 | 49585 | 46455 | 49544 |
| *SAR 60 cm kyr$^{-1}$ BD 10 cm* | 2866-2867 | 47725 | 44912 | 47775 | 2986-2987 | 49710 | 46556 | 49621 |

**Table 1.** The first downcore discrete-depth where "$^{14}$C-dead whole" foraminifera occur (i.e. $n_{dead} \geq 1$) for the various constant SAR and broken foraminifera scenarios discussed in Section 3 of this study. Also shown are the simulated median true ages, AMS $^{14}$C ages and median $^{14}$C calibrated ages corresponding to the discrete depth. The simulation analytical blank value is set at 46806 $^{14}$C yr BP (see Section 2.1), thus any single foraminifera with a $^{14}$C age older than that blank value are assumed "$^{14}$C-dead".

**Figure 1.** Overview of results of simulations using *Marine13* $\Delta^{14}$C involving multiple constant SAR scenarios (5, 10, 20, 40 and 60 cm kyr$^{-1}$) with constant BD of 10 cm, constant species abundance of 100% and 0% broken foraminifera. All discrete-depth results are plotted against their true median age on the x-axes. **(a)** The discrete-depth offset between mean AMS (i.e. laboratory) conventional $^{14}$C age and the idealised mean $^{14}$C age. **(b)** The discrete-depth offset between the true median age and the calibrated median age (i.e. that derived from the $^{14}$C dating and calibration process). **(c)** The discrete-depth difference between the calibrated highest posterior density (HPD) 95.4% age range (i.e that derived from the $^{14}$C dating and calibration process) and the true 95.4% age range of the sediment. **(d, e, f, g, h, i)** A visualisation of $^{14}$C calibration skill for select discrete-depth samples from various scenarios indicated on the figure panels. The blue histograms represent the actual single-foraminifera simulation output: on the x-axis the true age distribution of the single foraminifera (with the blue diamond corresponding to the median true age), and on the y-axis the corresponding true $^{14}$C age distribution of the single foraminifera (with the blue diamond corresponding to the mean $^{14}$C age of all individual foraminifera). All histograms are shown using 30 ($^{14}$C) year bins. The pink distributions represent the current state-of-the art in $^{14}$C dating. The pink normal distribution on the y-axis represents an AMS $^{14}$C determination carried out upon the single specimens, where the pink square corresponds to its mean. The pink probability distribution on the x-axis represents the calibrated age PDF arising from the calibration of the aforementioned AMS $^{14}$C determination using *Marine13* (Reimer et al, 2013) and *MatCal* (Lougheed and Obrochta, 2016), where the pink square represents the median calibrated age. Also shown, for reference, are the *Marine13* calibration curve 1σ (dark grey) and 2σ (light grey) confidence intervals.

**Figure 2.** An overview of residence time of single foraminifera within the active BD for the various simulation scenarios detailed in Fig. 1, i.e. with a constant BD of 10 cm and a SAR of **(a)** 5 cm kyr$^{-1}$ **(b)** 10 cm kyr$^{-1}$ **(c)** 20 cm kyr$^{-1}$ **(d)** 40 cm kyr$^{-1}$ **(e)** 60 cm kyr$^{-1}$.

**Figure 3.** Overview of results of simulations using Marine13 $\Delta^{14}C$ involving multiple constant SAR scenarios (5, 10, 20, 40 and 60 cm kyr$^{-1}$) with constant BD of 10 cm, constant species abundance of 100% and 10% broken foraminifera. All discrete-depth results are plotted against their true median age on the x-axes. **(a)** The discrete-depth offset between mean AMS (i.e. laboratory) conventional $^{14}C$ age and the idealised mean $^{14}C$ age. **(b)** The discrete-depth offset between the true median age and the calibrated median age (i.e. that derived from the $^{14}C$ dating and calibration process). **(c)** The discrete-depth difference between the calibrated highest posterior density (HPD) 95.4% age range (i.e that derived from the $^{14}C$ dating and calibration process) and the true 95.4% age range of the sediment. **(d, e, f, g, h, i)** A visualisation of $^{14}C$ calibration skill for select discrete-depth samples from various scenarios indicated on the figure panels. The blue histograms represent the actual single-foraminifera simulation output: on the x-axis the true age distribution of the single foraminifera (with the blue diamond corresponding to the median true age), and on the y-axis the corresponding true $^{14}C$ age distribution of the single foraminifera (with the blue diamond corresponding to the mean $^{14}C$ age of all individual foraminifera). All histograms are shown using 30 ($^{14}C$) year bins. The pink distributions represent the current state-of-the art in $^{14}C$ dating. The pink normal distribution on the y-axis represents an AMS $^{14}C$ determination carried out upon the single specimens, where the pink square corresponds to its mean. The pink probability distribution on the x-axis represents the calibrated age PDF arising from the calibration of the aforementioned AMS $^{14}C$ determination using *Marine13* (Reimer et al, 2013) and *MatCal* (Lougheed and Obrochta, 2016), where the pink square represents the median calibrated age. Also shown, for reference, are the *Marine13* calibration curve 1σ (dark grey) and 2σ (light grey) confidence intervals.

**Figure 4.** An estimation of the contribution of "$^{14}$C-dead" (i.e. activity below the analytical blank value) foraminifera to discrete-depth sample activity plotted against the apparent AMS $^{14}$C mean age of the discrete-depth sample. Based on the simulation scenarios detailed in Fig. 1 and Fig 3 with a constant BD of 10 cm and **(a)** SAR of 5 cm kyr$^{-1}$ and 0% broken foraminifera, **(b)** SAR of 5 cm kyr$^{-1}$ and 10% broken foraminifera, **(c)** SAR of 10 cm kyr$^{-1}$ and 0% broken foraminifera **(d)** SAR of 10 cm kyr$^{-1}$ and 10% broken foraminifera, **(e)** SAR of 20 cm kyr$^{-1}$ and 0% broken foraminifera, **(f)** SAR of 20 cm kyr$^{-1}$ and 10% broken foraminifera, **(g)** SAR of 40 cm kyr$^{-1}$ and 0% broken foraminifera, **(h)** SAR of 40 cm kyr$^{-1}$ and 10% broken foraminifera, **(i)** SAR of 60 cm kyr$^{-1}$ and 0% broken foraminifera, **(j)** SAR of 60 cm kyr$^{-1}$ and 10% broken foraminifera.

**Figure 5.** Four dynamic input scenarios (each with a unique colour) with constant $\Delta^{14}$C, each involving dynamic input for **(a)** SAR, **(b)** BD, **(c)** species abundance, **(d)** reservoir age ($\Delta$R). A constant broken foraminifera percentage of 10% is applied in all cases. **(e)** For each scenario, the resulting discrete-depth offset between mean AMS (i.e. laboratory) conventional $^{14}$C age and the idealised mean $^{14}$C age. **(f)** For each scenario, the discrete-depth offset between the true median age and the calibrated median age (i.e. that derived from the $^{14}$C dating and calibration process). **(g)** For each scenario, the difference between the calibrated highest posterior density (HPD) 95.4% age range (i.e that derived from the $^{14}$C dating and calibration process) and the true 95.4% age range of the sediment. **(h, i, j, k)** A visualisation of $^{14}$C calibration skill for select discrete-depth samples from various scenarios indicated on the figure panels. The blue histograms represent the actual single-foraminifera simulation output: on the x-axis the true age distribution of the single foraminifera (with the blue diamond corresponding to the median true age), and on the y-axis the corresponding true $^{14}$C age distribution of the single foraminifera (with the blue diamond corresponding to the mean $^{14}$C age of all individual foraminifera). All histograms are shown using 30 ($^{14}$C) year bins. The pink distributions represent the current state-of-the art in $^{14}$C dating. The pink normal distribution on the y-axis represents an AMS $^{14}$C determination carried out upon the single specimens, where the pink square corresponds to its mean. The pink probability distribution on the x-axis represents the calibrated age PDF arising from the calibration of the aforementioned AMS $^{14}$C determination using a custom-made calibration curve with constant $\Delta^{14}$C (see Section 4.0) and *MatCal* (Lougheed and Obrochta, 2016), where the pink square represents the median calibrated age. Also shown, for reference, are the calibration curve 1$\sigma$ (dark grey) and 2$\sigma$ (light grey) confidence intervals.

**Multiple constant SAR scenarios with Marine13 Δ$^{14}$C, constant BD of 10 cm, constant abundance of 100% and 0% broken foraminifera**

**Residence time in the active bioturbation depth**

(a) Constant SAR: 5 cm ka$^{-1}$
Constant BD: 10 cm ka$^{-1}$
Median residence: 2700 years
90th percentile

(b) Constant SAR: 10 cm ka$^{-1}$
Constant BD: 10 cm ka$^{-1}$
Median residence: 1370 years
90th percentile

(c) Constant SAR: 20 cm ka$^{-1}$
Constant BD: 10 cm ka$^{-1}$
Median residence: 690 years
90th percentile

(d) Constant SAR: 40 cm ka$^{-1}$
Constant BD: 10 cm ka$^{-1}$
Median residence: 350 years
90th percentile

(e) Constant SAR: 60 cm ka$^{-1}$
Constant BD: 10 cm ka$^{-1}$
Median residence: 230 years
90th percentile

Years

**Multiple constant SAR scenarios with Marine13 Δ$^{14}$C, constant BD of 10 cm, constant abundance of 100% and 10% broken foraminifera**

(a)

Legend:
- Constant SAR: 60 cm kyr$^{-1}$
- Constant SAR: 40 cm kyr$^{-1}$
- Constant SAR: 20 cm kyr$^{-1}$
- Constant SAR: 10 cm kyr$^{-1}$
- Constant SAR: 5 cm kyr$^{-1}$

(b)

(c)

(d) Constant SAR: 60 cm kyr$^{-1}$ / Constant BD: 10 cm / Discrete depth: 283-284 cm

(e) Constant SAR: 60 cm kyr$^{-1}$ / Constant BD: 10 cm / Discrete depth: 856-857 cm

(f) Constant SAR: 10 cm kyr$^{-1}$ / Constant BD: 10 cm / Discrete depth: 253-254 cm

(g) Constant SAR: 5 cm kyr$^{-1}$ / Constant BD: 10 cm / Discrete depth: 152-153 cm

(h) Constant SAR: 40 cm kyr$^{-1}$ / Constant BD: 10 cm / Discrete depth: 1529-1530 cm

(i) Constant SAR: 20 cm kyr$^{-1}$ / Constant BD: 10 cm / Discrete depth: 823-824 cm

Actual distribution of foraminifera

Distribution estimated using current $^{14}$C dating and calibration state-of-the-art

# Contribution of apparent "$^{14}$C-dead" foraminifera to AMS dates

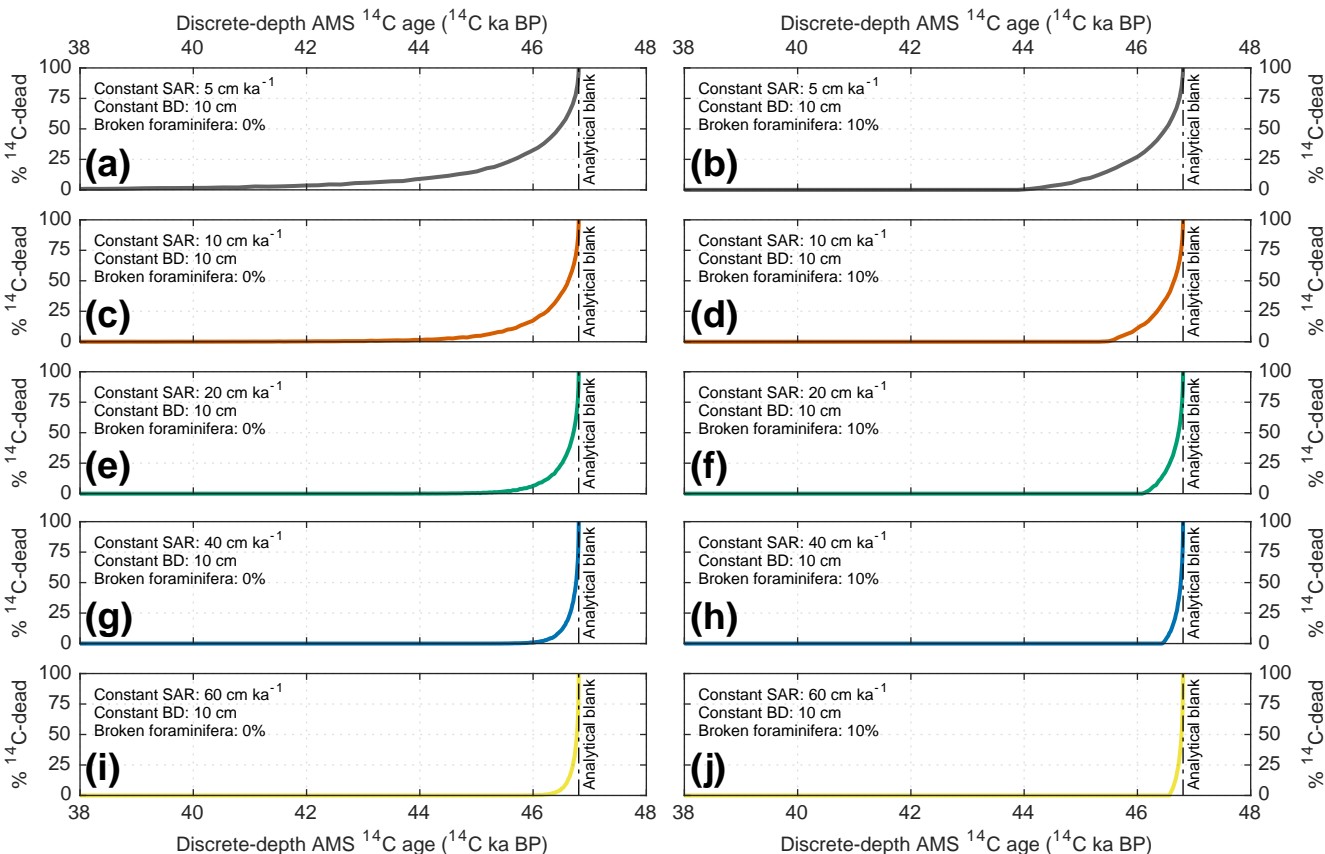

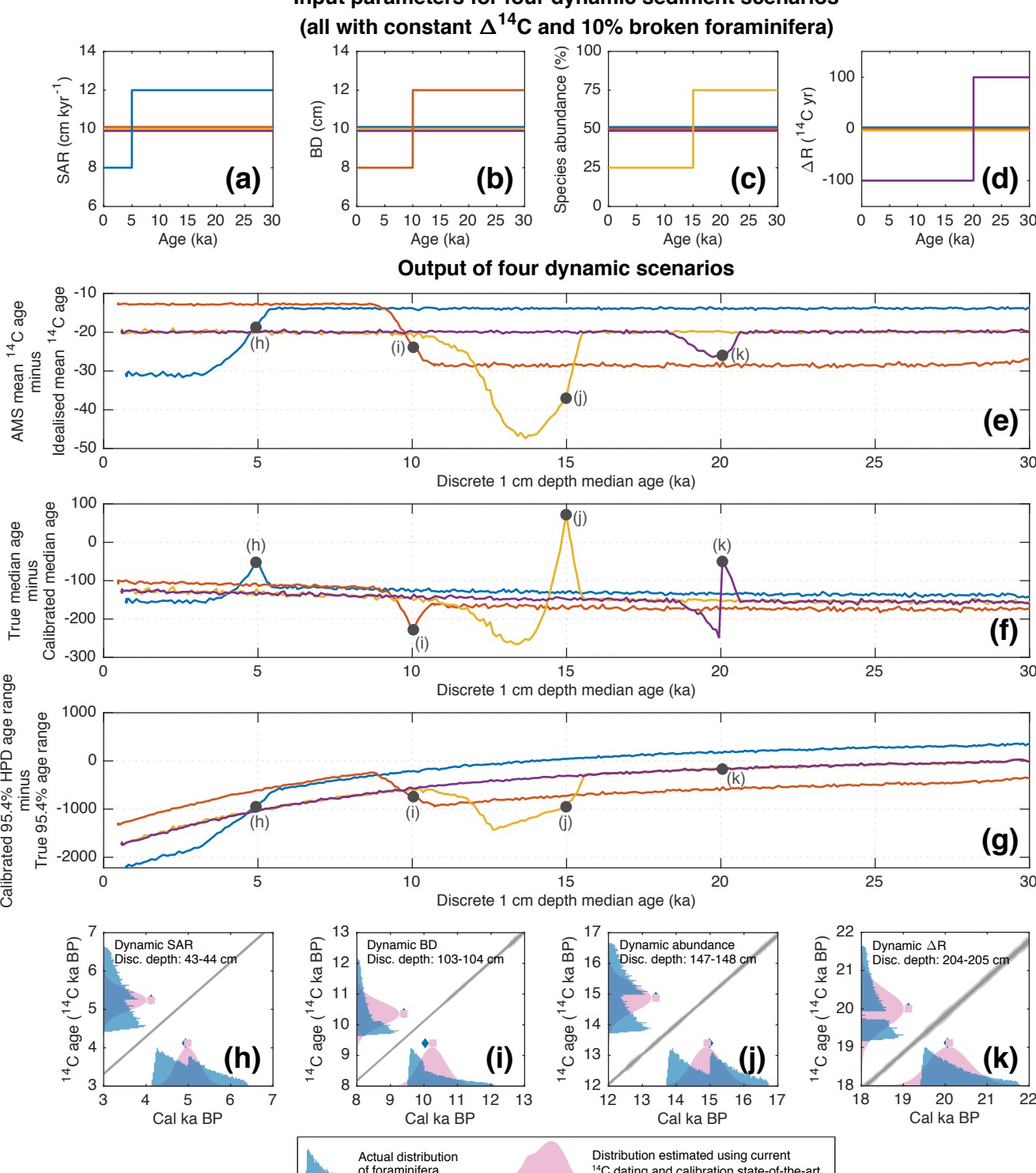

**Input parameters for four dynamic sediment scenarios (all with constant Δ¹⁴C and 10% broken foraminifera)**

(a) (b) (c) (d)

**Output of four dynamic scenarios**

(e) (f) (g)

(h) Dynamic SAR
Disc. depth: 43-44 cm

(i) Dynamic BD
Disc. depth: 103-104 cm

(j) Dynamic abundance
Disc. depth: 147-148 cm

(k) Dynamic ΔR
Disc. depth: 204-205 cm

Actual distribution of foraminifera

Distribution estimated using current ¹⁴C dating and calibration state-of-the-art