# Peer review of "Re-evaluating 14C dating accuracy in deep-sea sediment archives."

_Geochronology, 2019_

## Referee Comment (RC1) · Anonymous Referee #1 · 4 Nov 2019

Lougheed et al. present a number of sensitivity studies on the influence of bioturbation on marine sedimentary proxy records and assess how bioturbation may bias chronological age models derived from foraminiferal radiocarbon dates. The authors specifically test the impact of bioturbation as a function of sedimentation rate, bioturbation mixed-layer depth as well as abundance variations and fragmentation of foraminifera.

In my view, this work is important and highly useful that complements and expands existing numerical tools for assessing bioturbation in marine sediments (e.g., Trauth, 2013; Dolman and Laepple, 2018) and that will help to assess possible bioturbation biases of paleo-climate proxy records, which has increasingly been found and highlighted in the last years (e.g., Costa et al., 2017; Ausín et al., 2019). However, the impact of bioturbation on marine sediment records has been known for decades, and

similarly, bioturbation models are around for a long time, as the study acknowledges. The work of Lougheed et al. may therefore seem marginal, but I consider the implementation of single-specimen simulations and the direct link to 14C calibration tools important.

The authors perform their sensitivity analyses with the new tool SEAMUS that the first author developed and presents in an accompanying paper in Geoscientific Model Developments currently in review (https://doi.org/10.5194/gmd-2019-155). This work is another useful tool for the paleoclimate community by the first author that follows a number of existing applications broadly related to the same topic, e.g., MatCal (Lougheed and Obrochta, 2016). MatCal is an integral part of SEAMUS. The paper is well structured and is written in a clear way. I therefore believe that this paper is of great interest and use to the geochronology- and paleoclimate community, and I therefore recommend publication with minor revisions.

Along with one comment of major concern, I outline some minor points of criticism below in the hope that the authors find these useful in making the study further accessible to the readership. Although I have read the accompanying paper in Geoscientific Model Developments, I haven't had the time to run SEAMUS with study-independent scenarios. I assume that SEAMUS was intensively tested and run by reviewers who have assessed the accompanying paper.

Main criticism: Most of the paper deals with best-case scenarios, in which sedimentation rate, fragmentation and bioturbation depth are held constant. This rarely applies to marine sediment cores, as acknowledged. SEAMUS offers the opportunity to test the impact of bioturbation on chronological models under a variety of scenarios with transient changes of input variables, which would more closely represent marine sediment cores. This is addressed in section 4.0, but I must say that I expected a more detailed assessment of such non-ideal and more realistic scenarios (essentially more than half a page of discussion and analyses). I hence recommend that the authors run three different simulations to give it the attention it deserves and to further highlight the power

of their approach: One with the suggested step-like function in sedimentation rate, one with the suggested change in species abundance, and one with both changes. This would help to understand what process impacts the multi-modal character of the true age population the most and in what way. Furthermore, do both processes contribution equally to the offset between the AMS mean 14C age and the idealized mean 14C age, and the calibrated median age that is derived from the AMS mean age? These aspects should be discussed, in my view.

Minor comments: One key observation is that mean AMS 14C ages are generally younger than idealized mean 14C ages. The authors attribute this to the isotope mass balance effect. However, to me it seems that it must also be to some extent linked to the effect of bioturbation itself, essentially the character of the exponential pdf for true age (how extended and pointy it is), combined with character of the atmospheric calibration. I wonder whether the authors can elaborate on additional processes in case they think these may be important, or specify whether they think the isotope mass balance effect is the main driver of the observed offsets.

"Perfect" simulated sediment archive scenarios: I would argue that a "perfect" sediment core is one without and very little bioturbation (e.g., BD=1cm with high SR>10 cm). I think it is worth to rephrase to "most favorable" or "best-case".

Section 2.2. It needs to be specified what fraction of the total number of foraminifera has been picked by the simulations, maybe simply by inserting "with a fraction set by the operator". I would expect that this fraction matters for the magnitude of offset between AMS mean 14C age and whole-sample/idealized 14C age. I think it is worth highlighting that this fraction in reality changes as the abundance of (well-preserved) foraminifera changes in sediment cores, and that this would affect the observations (in what way?).

Line 155: it is unclear what is meant with "samples nearing the blank value". I would argue that samples with a 14C age of 45 kyr BP have a much larger analytical uncertainty

(>200 yr). Please clarify.

Line 204-207: Do you mean that the AMS 14C age diverges from the idealized mean 14C age? I am confused here that an offset diverges from something. Please clarify.

Line 208: I think it would help to specify the following "for each discrete depth FROM THE ARTIFICIALLY PICKED SAMPLES", because simulated AMS mean and idealized mean 14C age are introduced here for the first time.

Line 250: Rephrase to express that the true median age is lower than the calibrated mean age, as "offset" can be both negative and positive.

Line 258: I would recommend to add "throughout the sedimentation history of a sediment archive, MUCH MORE EFFICIENTLY THAN YOUNG FORAMINIFERA DOWNWARDS"

Line 286: I agree that a sample is 14C-dead when it is older than the blank, but shouldn't it be declared 14C-dead when within 2-sigma of the blank too (Stuiver and Polach, 1977). If the authors agree, I'd recommend clarification of this sentence.

Line 323-326: I am surprised that the offsets around 11 kyr are emphasized, although they occur throughout the simulation, with a similar or larger magnitude. Please revise.

Typos: Line 176: remove "it that" Line 215: remove "much" Line 223: space missing between idealized and mean Line 230: remove "of" Line 280: add "of" at the end of line Line 297: remove "4" Line 302: insert "that" after "this effect is" Line 334: remove both "/" Figure caption 1, 3 and 5: change "i.e" into "i.e." or "i.e.," ; 14 in "14C" should be superscript; I am also somewhat confused by the (what seems to me an inconsistent) use of "mean 14C age" and "idealized", in particular with regards to how panel a is labelled. Shouldn't it say in the last sentences: "on the y-axis the corresponding 14C age distribution of the single specimens (with the blue diamond corresponding to the AMS mean 14C age). [...] The pink normal distribution on the y-axis represents the IDEALISED 14C AGE UNCERTAINTY of the single specimens, where the pink square

corresponds to the expected IDEALISED MEAN 14C age." This may seem marginal but it would help to keep the terminology consistent across figure caption and figure labels.

References: Ausín, B., Haghipour, N., Wacker, L., Voelker, A.H.L., 2019. Radiocarbon Age Offsets Between Two Surface Dwelling Planktonic Foraminifera Species During Abrupt Climate Events in the SW Iberian Margin Paleoceanography and Paleoclimatology. Paleoceanogr. Paleoclimatology 34, 63–78. https://doi.org/10.1029/2018PA003490 Costa, K.M., McManus, J.F., Anderson, R.F., 2017. Radiocarbon and stable isotope evidence for changes in sediment mixing in the North Pacific over the past 30 kyr. Radiocarbon 1–23. https://doi.org/10.1017/RDC.2017.91 Dolman, A.M., Laepple, T., 2018. Sedproxy: A forward model for sediment-archived climate proxies. Clim. Past 14 (12), 1851–1868. https://doi.org/10.5194/cp-14-1851-2018 Lougheed, B.C., Obrochta, S.P., 2016. MatCal: Open Source Bayesian 14C Age Calibration in Matlab. J. Open Res. Softw. 4, 1–4. https://doi.org/10.5334/jors.130 Stuiver, M., Polach, H.A., 1977. Discussion: Reporting of 14C data. Radiocarbon 19 (03), 355–363. https://doi.org/10.1017/S0033822200003672 Trauth, M.H., 2013. TURBO2: A MATLAB simulation to study the effects of bioturbation on paleoceanographic time series. Comput. Geosci. 61, 1–10. https://doi.org/10.1016/j.cageo.2013.05.003

---

## Author Comment (AC1) · 17 Nov 2019

We would like to thank Referee #1 for their detailed and helpful review of our manuscript. We especially appreciate that the referee has volunteered additional time to read the accompanying model description paper that is published in another journal, thus further enhancing the quality of the review.

We respond to the referee's points below, whereby the referee's comments are indented.

> However, the impact of bioturbation on marine sediment records has been known for decades, and similarly, bioturbation models are around for a long time, as the study acknowledges. The work of Lougheed et al. may therefore seem marginal, but I consider the implementation of single-specimen simulations and the direct link to 14C calibration tools important.

We think this description by the referee sums up the manuscript well. Bioturbation is a well-known, but nevertheless often disregarded and misunderstood phenomenon. Its specific effect upon (calibrated) $^{14}C$ chronologies in deep-sea sediment cores has not yet not been systematically quantified, probably due to 20th century limitations in computing power.

> Although I have read the accompanying paper in Geoscientific Model Developments, I haven't had the time to run SEAMUS with study-independent scenarios. I assume that SEAMUS was intensively tested and run by reviewers who have assessed the accompanying paper.

Yes, the model description manuscript has now been positively assessed by two referees. Of course, no model can be conclusively guaranteed as bug-free, but we note that the output of the single-foram enabled SEAMUS model was tested in the model description paper against a well-established 'traditional' bioturbation model that considers only the mean, downcore signal (TURBO2), and the downcore mean signal output of both was found to be in virtually perfect agreement.

> Main criticism: Most of the paper deals with best-case scenarios, in which sedimentation rate, fragmentation and bioturbation depth are held constant. This rarely applies to marine sediment cores, as acknowledged. SEAMUS offers the opportunity to test the impact of bioturbation on chronological models under a variety of scenarios with transient changes of input variables, which would more closely represent marine sediment cores.

In this manuscript we chiefly seek to concentrate upon an effect that has, as far as we know, not previously been considered nor mentioned in the literature: namely age-depth artefacts that can arise as a result of the mischaracterisation of pooled foraminifera-based $^{14}C$ determinations as a normal distribution, coupled to the subsequent amplification of this mischaracterisation during the calibration process. In order to study this effect in isolation, it is necessary to construct model environments whereby all input variables except for $\Delta^{14}C$ are kept constant. We will make our aims clearer in the abstract and introduction of the final manuscript.

Indeed, there are many temporally dynamic variables which affect the distribution of single foraminifera $^{14}C$ values within pooled sediment core samples, including sedimentation rate, bioturbation depth, species abundance changes, local reservoir age, global $\Delta^{14}C$, foraminifera vulnerability to dissolution/breakage, number of foraminifera picked per sample, etc. The number of figures that we could have generated is, therefore, practically infinite, as the referee will appreciate. We do agree with the referee that it would be possible to add one or two more 'dynamic' scenarios for the benefit of the reader.

> I hence recommend that the authors run three different simulations to give it the attention it deserves and to further highlight the power of their approach: One with the suggested step-like function in sedimentation rate, one with the suggested change in species abundance, and one with both changes.

> This would help to understand what process impacts the multi-modal character of the true age population the most and in what way.

These two additional simulations would indeed add useful information for the reader when it comes to judging the effect of dynamic input variables and the creation of further age-depth artefacts. We did not want to make too many figures, hence we only included a dynamic simulation with a step in sedimentation rate. However, on second thought, it may be possible to include the helpful suggestion of the referee in the same figure by adding lines with different colours for each scenario.

> One key observation is that mean AMS 14C ages are generally younger than idealized mean 14C ages. The authors attribute this to the isotope mass balance effect. However, to me it seems that it must also be to some extent linked to the effect of bioturbation itself, essentially the character of the exponential pdf for true age (how extended and pointy it is), combined with character of the atmospheric calibration.

Whenever there is sample age heterogeneity of any type, the AMS $^{14}$C age (the average of all $F^{14}$C values of the foraminifera in the sample, which is subsequently converted to 14C yr) will always be younger than the idealised mean $^{14}$C age. This effect is because the $F^{14}$C scale (or pMC scale) is exponential vs time and is, therefore, an accurate reflection of the isotope mass balance of radioactive $^{14}$C, whereas $^{14}$C years are linear vs time. The offset between the AMS $^{14}$C age and the idealised $^{14}$C age increases as the radiocarbon heterogeneity contained within the sample increases – this can be attributed to e.g. reduced sedimentation rate, periods of dynamic $\Delta^{14}$C etc, as the referee notes. We did discuss all these factors in depth in lines 200 to 225. We will try to improve the structure of that text in the final version.

> "Perfect" simulated sediment archive scenarios: I would argue that a "perfect" sediment core is one without and very little bioturbation (e.g., BD=1cm with high SR>10 cm). I think it is worth to rephrase to "most favorable" or "best-case".

Good point, will rephrase for clarity.

> Section 2.2. It needs to be specified what fraction of the total number of foraminifera has been picked by the simulations, maybe simply by inserting "with a fraction set by the operator".

In Section 2.2 we describe the methods in general, in 3.0 we describe the different runs, where the exact settings used (100% of forams picked or ignoring the 10% oldest forams) are stated. We will make this clearer in the final manuscript, as it is indeed jarring at the moment when reading the manuscript from beginning to end.

> I would expect that this fraction matters for the magnitude of offset between AMS mean 14C age and whole-sample/idealized 14C age. I think it is worth highlighting that this fraction in reality changes as the abundance of (well-preserved) foraminifera changes in sediment cores, and that this would affect the observations (in what way?).

Regarding the first sentence of the comment: this effect can be clearly seen in the results, when one compares Fig. 1a (0% broken forams) and Fig. 2a (10% broken forams) and sees that the offset is systematically less in the case of the latter. We may have failed to mentioned this in the text, and will check to make sure we do when preparing the final version of the manuscript. Regarding the second sentence of the comment: Yes, in practice the fraction of broken foraminifera can of course change through time in the field as a function of dissolution, foraminifera ecology, bioturabtion depth, sedimentation rate. This issue is alluded to but should indeed be made clearer.

Line 155: it is unclear what is meant with "samples nearing the blank value". I would argue that samples with a 14C age of 45 kyr BP have a much larger analytical uncertainty. Please clarify.

Yes, "samples nearing the blank value" is indeed vague text. We will include the following in the final manuscript for clarity: " Specifically, when assigning measurement errors to synthetic AMS determinations, a $^{14}$C determination of 1.0 F$^{14}$C is assumed to have an error of 30 $^{14}$C yr, and a determination with the F$^{14}$C value $e^{(\text{blankvalue-1})/-8033}$ (i.e. one $^{14}$C yr younger than the blank value) is assumed to have an error of 200 $^{14}$C yr. Errors (in $^{14}$C yr) for intermediate dates are linearly interpolated to F$^{14}$C. "

Regarding the choice of error value: 200 $^{14}$C yr can indeed be considered an optimistic (but not unrealistic) measurement error for large (>1 mg) carbonate samples of ~46000 $^{14}$C yr BP measured as graphite targets on the latest AMS systems. However, in our simulations we assume "best-case" conditions, thus also best case conditions for AMS measurement.

The remaining, more minor comments by the referee will also be addressed in the manuscript.

We'd like to thank the referee again for their extensive and helpful review.

---

## Referee Comment (RC2) · Anonymous Referee #2 · 20 Jan 2020

Lougheed et al. address the influence of bioturbation in ocean sediments on the accuracy of sediment ages determined by 14C dating. Accurate ages are relevant for global correlation of sediment records and thus for a better understanding of the interactions of oceanography, climate, and the carbon cycle in the past and future. As such the subject of the paper is highly relevant. The modelling approach using the established SEAMUS model is clearly described and the many processes that can influence the 14C concentration of a sample of foraminifera picked from a discrete sediment layer are indicated.

The paper focuses on the calculation of the age spectrum of foraminifera in a discrete sediment section resulting from bioturbation and demonstrates that the average age of the individual foraminifera generally will differ from the age derived from a measure-
ment of their combined 14C content. In practical sediment dating, the aim is, generally, to establish the time of deposition of the particular sediment section by determining the 14C content of planktic foraminifera deposited coevally. Thus the demonstration that bioturbation may significantly affect the 14C content of planktic foraminifera in a sediment section (5.0 Conclusion) does not directly contribute to a better age determination of ocean sediments (6.0 Outlook).

Of practical use would be modelling of the difference between the average planktic 14C content and the planktic 14C at time of deposition of a section. Comparing this difference with the uncertainties of the 14C measurement and the estimation of the original planktic 14C content, would show where bioturbation influence may be negligible, where a correction should be attempted, and the added age uncertainty due to bioturbation. It should be noted that the model results presented have been obtained under very idealized conditions, as clearly stated in the model assumptions. To demonstrate the value of SEAMUS in the real world it would be good to see results for common sized mono-specific foram samples (0.1 to 1.0 mg C, $\sim$30 to 300 shells) selected from a sediment section and modelled with sedimentation rates and species abundance as well as local surface reservoir age varying over time according to a realistic local scenario. The uncertainty of the measured age, depending on sample 14C concentration, background, and surface reservoir age uncertainty, may vary from 0.2% for very young to several % for old (>30 ka) samples.

A critical issue to be addressed for the use of single forams, that now may become possible as mentioned in Outlook, is the variability in the isotopic signal of individual coeval foraminifera. During the lifetime of a single foram the 14C concentration of the water surrounding the foram may vary due to varying ocean-atmosphere exchange, turbulent mixing with deeper layers, planktic bloom, and change in depth of the foram as it ages. The natural spread in 14C concentration, $\delta$13C, and $\delta$18O in a population of contemporaneous foraminifera needs to be determined and compared with the magnitude of the paleoclimatic signals expected to see what Information may be obtained.

To a lesser degree this individual variability also needs to be considered in deciding to what extent a finite number of shells is representative for conditions at the ocean surface. 300 foraminifera may be representative, for a sample of 30 shells it may be an uncertainty factor.

In conclusion, I agree with the authors that bioturbation needs to be considered when interpreting 14C ages obtained on planktic foraminifera samples but I find the present paper too far removed from reality to be published in Geochronology. In its present form it seems more appropriate for a modelling-oriented journal.

I would like to encourage the authors to invest a bit more time in this interesting work by running their model under (more) realistic conditions, as mentioned above, and comparing the results for the 14C age that will be measured with the quantity sought in sediment dating, i.e. the time of deposition of the sediment section and the bulk of the foraminifera in it. Such a paper would be highly suitable for Geochronology and the 14C dating of deep-sea sediment archives.

Specific comments:

Line 44: True difference in age is not the only possible cause of 14C age heterogeneity. Other causes as listed in lines 83-87 also come into play.

Line 77: "14C history of the Earth" is too general. It is better to separate the atmospheric 14C history, which is largely global, from the oceanic 14C history, which is strongly local.

Line 90: The discussion does not differentiate between the probability distribution of the measured 14C concentration and that of the related 14C age although the latter follows nonlinearly, via e-log, from the first, which for old samples has significant consequences . Line 118: 10 4 specimens repressents ideal conditions compared with 30 to 300 foraminifera selected from the population of the 1-cm section.

Line 121: the primary parameter is F14C, an apparent 14C age follows from it. Although Marine20 is quite different from Marine13 beyond 14 ka, this is for the discussion of the technique, at present, not important.

Line 130: There seems to be confusion on the meaning of blank value. 14C convention is that only definitive 14C values (measured minus background) that exceed twice their uncertainty should be given. This does not mean that foraminifera older than this limiting age/concentration have all the same 14C concentration. They don't; their 14C content keeps decreasing but we can no longer reliably measure it. Thus assigning a constant 14C concentration to all older forams reduces the calculated effect of upward mixing of old forams in deep core sections.

Line 155: Near the blank value the age uncertainty will be asymmetric and generally significantly larger than 200 years because not only the uncertainty in the measured 14C but also that of the blank to be subtracted has to be considered.

Line 209: Note that one is usually seeking the time of deposition of the section and thus the 14C age of the foraminifera raining down at time of deposition. The bias of measured age relative to this will be towards older.

Line 245: Is the second decimal in 95.45 % relevant? Usually only one is given.

Line 287: Are the artefactually young 14C ages the result of assigning a constant "blank" 14C concentration to older foraminifera (see line 130 above)? Modelling could be changed.

Line 297: 1% contribution of 14C free carbon is equivalent with 1 % decay, meaning 80 years too old. In the age range mentioned here, this is well below the measurement uncertainty (i.e. fortunately negligible).

Line 355: The statement that considering bioturbation could improve dating accuracy certainly is true. More realistic modelling is, however, needed to demonstrate the potential of SEAMUS to produce significant improvements.

The authors should check for text duplications.

---

## Author Comment (AC2) · 28 Jan 2020

Reply to Referee #2. Our replies are indented.

**Main referee comments:**

Lougheed et al. address the influence of bioturbation in ocean sediments on the accuracy of sediment ages determined by 14C dating. Accurate ages are relevant for global correlation of sediment records and thus for a better understanding of the interactions of oceanography, climate, and the carbon cycle in the past and future. As such the subject of the paper is highly relevant. The modelling approach using the established SEAMUS model is clearly described and the many processes that can influence the 14C concentration of a sample of foraminifera picked from a discrete sediment layer are indicated.

> We thank Referee #2 for their kind words and the time they have taken to provide their insight on our manuscript.

It should be noted that the model results presented have been obtained under very idealized conditions, as clearly stated in the model assumptions. To demonstrate the value of SEAMUS in the real world it would be good to see results for common sized mono-specific foram samples (0.1 to 1.0 mg C, ∼30 to 300 shells) selected from a sediment section and modelled with sedimentation rates and species abundance as well as local surface reservoir age varying over time according to a realistic local scenario. [...] In conclusion, I agree with the authors that bioturbation needs to be considered when interpreting 14C ages obtained on planktic foraminifera samples but I find the present paper too far removed from reality to be published in Geochronology. In its present form it seems more appropriate for a modelling-oriented journal.

I would like to encourage the authors to invest a bit more time in this interesting work by running their model under (more) realistic conditions, as mentioned above, and comparing the results for the $^{14}$C age that will be measured with the quantity sought in sediment dating, i.e. the time of deposition of the sediment section and the bulk of the foraminifera in it. Such a paper would be highly suitable for Geochronology and the 14C dating of deep-sea sediment archives.

> We respectfully disagree with the above comments. One of the major advantages of computer modelling analysis of complex systems is that a model can be used as an investigative tool to analyse the influence of an isolated parameter upon an entire system. Consider for a moment some of the classic modelling papers which have allowed us to better understand the Earth's climate system by keeping all climate parameters constant except for the one of interest (e.g. $pCO_2$). One could equally argue that such classic modelling papers are "removed from reality", but it is precisely for that reason that those papers are of great importance and why they provide valuable new insights into the functioning of a complex system.

> We also follow such a classic modelling approach. In our case we have taken advantage of computer modelling to construct an ideal experimental design whereby we can evaluate how the current $^{14}$C state-of-the-art would work in the case of best-case conditions. Since these best-case conditions do not exist in reality, a computer modelling environment can uniquely be used to create such a best-case scenario, whereby the long-established and accepted understanding of bioturbation is incorporated into the model. This approach allows us to test the accuracy of the current $^{14}$C dating state-of-the-art applied to deep-sea sediments on the most fundamental level. If it is demonstratively shown that the current state-of-the-art functions sub-optimally even in best-case conditions, then very important questions are raised, which are highly relevant for the readership of *Geochronology*.

> Of course we agree that in the case of "real-world" deep-sea sediment all input parameters do indeed exhibit constant temporal variation (which SEAMUS is capable of modelling), thus

increasing the degrees of freedom of practical experimental design for real-world studies in the field, which subsequently places strong interpretive limitations on such real-world studies. If we were to increase the degrees of freedom of the experimental design in our model study to mimic real-world conditions and their associated interpretive limitations, we would be unable to provide any new insights that have not already been provided by field-based studies. A huge advantage of a computer model is the ability to overcome such limitations by constraining multiple input parameters, allowing for new insights into the fundamental functioning of a complex system to be found, which we believe we have done. For example, we have shown that even in the case of best-case conditions, the current [14]C dating state-of-the-art will produce systematic age-depth artefacts which are a result a misrepresentation of the [14]C age distribution of the sample by the current state-of-the-art, combined with the mixing of single foraminifera from differing periods of the Earth's history.

Finally, we note that the referee did not actually find a significant fault with the modelling processes performed in our study.

**Other referee comments addressed below:**

A critical issue to be addressed for the use of single forams, that now may become possible as mentioned in Outlook, is the variability in the isotopic signal of individual coeval foraminifera. During the lifetime of a single foram the 14C concentration of the water surrounding the foram may vary due to varying ocean-atmosphere exchange, turbulent mixing with deeper layers, planktic bloom, and change in depth of the foram as it ages. The natural spread in 14C concentration, δ13C, and δ18O in a population of contemporaneous foraminifera needs to be determined and compared with the magnitude of the paleoclimatic signals expected to see what Information may be obtained. To a lesser degree this individual variability also needs to be considered in deciding to what extent a finite number of shells is representative for conditions at the ocean surface. 300 foraminifera may be representative, for a sample of 30 shells it may be an uncertainty factor.

These are all very interesting details which we have often pondered ourselves, but go beyond this current study. Our study aims to evaluate the current state-of-the-art of [14]C dating of deep-sea sediments, by testing how the current state-of-the art performs in best-case conditions in deep-sea sediment. Including many more variable parameters in our simulation would increase the degrees of freedom of the experimental design and preclude us from making useful interpretations. Hypothetical questions about the intra-shell [14]C variability of single foraminifera as they transition through water bodies during their lifetime are well beyond the scope of our study. We refer the referee to many studies about changes in the [14]C activity of marine waters, i.e. studies about spatio-temporal changes in marine reservoir age.

In practical sediment dating, the aim is, generally, to establish the time of deposition of the particular sediment section by determining the 14C content of planktic foraminifera deposited coevally.

Respectfully, the oft-applied assumption that foraminifera retrieved from a given centimetre(s) thick interval of a deep-sea sediment archive were deposited at the sea floor coevally (*i.e.* at the same time) is one of the main assumptions we seek to re-evaluate in this manuscript.

Thus the demonstration that bioturbation may significantly affect the 14C content of planktic foraminifera in a sediment section (5.0 Conclusion) does not directly contribute to a better age determination of ocean sediments (6.0 Outlook).

Section 5.0 describes previously undescribed challenges to [14]C dating that we have highlighted in our study. Section 6.0 proposes future methods with which these challenges can be quantified

when calibrating multi-specimen samples and/or overcome by using new analytical methods. We will make clearer that the Outlook section is describing potential future studies/remedies and not resolutions that we have produced in this study.

Of practical use would be modelling of the difference between the average planktic 14C content and the planktic 14C at time of deposition of a section.

We're not sure if we fully understand this comment. The $^{14}$C activities (relative to 1950 CE) assigned to the simulated planktonic foraminifera do not change due to deposition.

Line 44: True difference in age is not the only possible cause of 14C age heterogeneity. Other causes as listed in lines 83-87 also come into play.

Indeed, as we mention in lines 83-87, as the referee points out. We will attempt to better foreshadow this in the text for the benefit of the reader.

Line 77: "14C history of the Earth" is too general. It is better to separate the atmospheric 14C history, which is largely global, from the oceanic 14C history, which isstrongly local.

Agreed, we will make this sentence more specific.

Line 90: The discussion does not differentiate between the probability distribution of the measured 14C concentration and that of the related 14C age although the latter follows nonlinearly, via e-log, from the first, which for old samples has significant consequences.

The $^{14}$C calibration software embedded in SEAMUS is MatCal (Lougheed and Obrochta, 2016), which calibrates $^{14}$C determinations in $F^{14}$C space, so there is no trouble with calibrating older samples in the case of SEAMUS. We should indeed mention this somewhere for the benefit of the reader!

Line 118: $10^4$ specimens repressents ideal conditions compared with 30 to 300 foraminifera selected from the population of the 1-cm section.

We aim to investigate how well the current $^{14}$C state-of-the-art performs under ideal conditions, and adding further degrees of freedom to our experimental design by including additional noise created by small and/or variable sample sizes would impede our aim.

Line 121: the primary parameter is F14C, an apparent 14C age follows from it. Although Marine20 is quite different from Marine13 beyond 14 ka, this is for the discussion of the technique, at present, not important.

Each single foraminifera within the simulation is first assigned a $^{14}$C activity in conventional $^{14}$C age, which is subsequently converted to $F^{14}$C. The reason for this approach is that we use the *Marine13* curve to assign $^{14}$C activity, which is published by the IntCal group, who report activity as conventional $^{14}$C age (i.e. in the downloadable "*Marine13.14c*" file). When it comes to assigning $^{14}$C activity to single foraminifera, the two units ($^{14}$C age and $F^{14}$C) are readily convertible via a simple formula, so there are no consequences for our study.

*Marine20* may indeed be different from *Marine13* for older parts due to the improved and extended Hulu Cave record, which we allude to in lines 225 to 230.

Line 130: There seems to be confusion on the meaning of blank value. 14C convention is that only definitive 14C values (measured minus background) that exceed twice their uncertainty should be given. This does not mean that foraminifera older than this limiting age/concentration have all the same 14C concentration. They don't; their 14C content keeps decreasing but we can no longer reliably measure it.

We thank the referee for making us aware that we need to describe this part of the method in a more clear way. We are of course aware that $^{14}$C activity continues to decrease with time according to the principles of radioactive decay. The reason we assign, within the simulation, all foraminifera older than the blank value the same $^{14}$C activity (in F$^{14}$C) as the blank value itself, is so that we can simulate the practical $^{14}$C dating (AMS analysis) upon said foraminifera, i.e. the aim of our study. In our simulation the analytical blank value is set to 46806 $^{14}$C yr BP (assigned as 0.0029477 F$^{14}$C). This represents the signal that older foraminifera will contribute to the measurement process in practice, irrespective of their age. So, for example, for a given virtual sample containing 1000 foraminifera of between 85,000 and 90,000 years old, we would calculate the virtual AMS date by taking the mean of all their individually assigned F$^{14}$C values (the 'blank' value in this case), which would result in an AMS determination with a mean value of 0.0029477 F$^{14}$C, analogous to a real-world AMS determination of such material.

Line 155: Near the blank value the age uncertainty will be asymmetric and generally significantly larger than 200 years because not only the uncertainty in the measured14C but also that of the blank to be subtracted has to be considered.

As mentioned in a reply to a previous comment (Line 90), $^{14}$C dates within the simulation are calibrated in F$^{14}$C space using MatCal (Lougheed and Obrochta, 2016), so are not affected by such asymmetries.

As we stated to Referee #1, we are running a 'best-case' model evaluation of $^{14}$C dating, including best-case AMS analysis with a very optimistic uncertainty of 200 $^{14}$C yr for very old dates. We can re-run the simulations with an uncertainty of  ~700 $^{14}$C yr for very old dates, although it would have little bearing on our main conclusions.

Line 209: Note that one is usually seeking the time of deposition of the section and thus the 14C age of the foraminifera raining down at time of deposition. The bias of measured age relative to this will be towards older.

We don't fully understand this comment. In practical terms, $^{14}$C is used in palaeoceanography to determine the average (calibrated) age of the foraminifera contained within an interval of retrieved sediment core, so that an accurate age can be assigned to whatever climate/oceanography proxy was also retrieved from foraminifera contained within the same interval. Is the referee commenting on the transit time of a dead foraminifera from the ocean water surface to the seafloor? In that case the transit time is mere weeks, see following study: doi:10.1038/ncomms7521

Line 245: Is the second decimal in 95.45 % relevant? Usually only one is given.

There is actually no reason why we give a second number after the decimal point. We are unaware of any decimal convention for reporting HPD intervals of calibrated ages. We can change to one decimal point if necessary, and considering that the calibrated ages are rounded to one year, two decimal places is probably indeed a bit optimistic anyway!

Line 287: Are the artefactually young 14C ages the result of assigning a constant "blank" 14C concentration to older foraminifera (see line 130 above)? Modelling could be changed.

We refer to our response to the comment for Line 130. Note that in the text we refer artefactually young "*measured* $^{14}$C age" and that the x-axis of Fig. 4 is labelled as "Discrete-depth *AMS* $^{14}$C age". Consider a sample containing 70% foraminifera that have a $^{14}$C activity greater than the blank value, and 30% foraminifera with a $^{14}$C activity less than the blank value. In our simulation, the latter 30% will simply be assigned the blank value as their $^{14}$C activity, thus biasing the mean

sample activity of the sample calculated within the simulation towards a too high activity (= younger $^{14}$C age). This is analogous to what would happen in a real-world laboratory determination of such a sample (what we seek to simulate in our study), because the laboratory cannot measure below the blank value. We will attempt to make this point clearer in the final version, because at the moment it does appear confusing to the reader.

Line 297: 1% contribution of 14C free carbon is equivalent with 1 % decay, meaning 80 years too old. In the age range mentioned here, this is well below the measurement uncertainty (i.e. fortunately negligible).

For other % contributions we refer the referee to Fig. 4.

Line 355: The statement that considering bioturbation could improve dating accuracy certainly is true. More realistic modelling is, however, needed to demonstrate the potential of SEAMUS to produce significant improvements.

We refer to the rest of our reply.

Thanks again to the referee for taking the time to review the manuscript and providing their insight.

---

## Author Comment (AC3) · 3 Feb 2020

Dear Dr. Hajdas,

Thank you for your interest in our manuscript and for sending it out for review and discussion. Thanks also to the referees for taking their time to provide their insight into our work, which will help to improve the manuscript.

We have responded to both referees directly in separate replies and sum up our replies in a final author comment here.

Both Referee #1 and Referee #2 found that our model approach correctly incorporated the processes of sedimentation, bioturbation, $\Delta$14C, AMS measurement and the calibration process. Referee #2 considers the manuscript unsuitable for Geochronology,

due to our best-case scenario simulations not being a exact imitation of field condi-
tions, where such best-case scenarios do not exist. As we detailed in the manuscript
and further elaborated in the response to Referee #2, our best-case scenario approach
intentionally does not mimic reality, thus allowing us to construct a classic experimental
design whereby many input variables are kept constant, thereby testing the accuracy
of the current 14C dating state-of-the-art applied to deep-sea sediments at the most
fundamental level, in a way that is not possible in the field. We think that such a study
is inherently interesting for the readership of Geochronology. We will strive to improve
the clarity of our reasoning in an updated version of the manuscript.

Following the referee comments and our replies, we propose the following two main
action points for an updated manuscript:

(1) The main request of Referee #1 is that we include dynamic scenarios for other
variables in Figure 5. As mentioned in the reply to Referee # 1, we will include this in
Figure 5 in the form independent simulations with independent stepwise changes for
other variables (such as abundance, reservoir age, etc). This will allow the reader to
separately judge the influence of different drivers upon the 14C dating process.

(2) Both Referees #1 and #2 suggested that the simulated measurement uncertainty
of $\pm200$ 14C yr for very old 14C samples close to the blank level was overly optimistic.
We note that our study attempts to simulate best-case scenarios, including for mea-
surement. However, I have since been in contact with multiple laboratories regarding
the theoretical best-case measurement uncertainty in the case of minimal contamina-
tion, high-quality blanks, et cetera. We can conclude that while a best-case scenario
measurement error of $\pm200$ 14C yr for very old samples is in theory possible, it might
not be applicable to foraminifera samples, which are more susceptible to contamination
than, e.g., bone or wood samples. We will re-run the simulations with a more suitable
theoretical best-case value (e.g. $\pm500$ 14C yr) for completeness. The outcome of our
study will not be affected.

[Figure]

The other helpful points of the referees pertaining to better communication of certain concepts and clarification of methods will also of course be implemented, as detailed in the responses to the referees.

We thank you again for your interest in our manuscript and await your decision on how to proceed, including any suggestions you may have for the improvement of the manuscript.

On behalf of the authors,

Kind regards,

Bryan Lougheed
* * *

---

## Author Response (AR1)

Uppsala, Sweden
March 5, 2020

Dear Dr. Irka Hajdas,

Thank you for considering to allow us to submit a revised version of our manuscript and thanks to the referees for their input which has helped to improve the manuscript. Please find attached our updated manuscript, where we have completed the minor revisions that have been requested.

Overview of changes:

Figure 5 now shows four dynamic sediment scenarios with independent changes for the following dynamic parameters: **(1)** sediment accumulation rate (SAR), **(2)** bioturbation depth (BD), **(3)** species abundance; **(4)** reservoir age. These particular dynamic sediment scenarios use constant $\Delta^{14}C$ (instead of Marine13), so that the reader can independently judge the effect of the dynamic input parameters.

The introduction has been rewritten to incorporate "Background and Rationale" and "Experimental Design" subsections which help lay out the rationale behind the study and the unique strength in using 'best-case' scenarios for the goal of our study. This should help the readers to better understand the value of our modelling study.

After further feedback, legends have been added to the calibration plots in Fig 1, Fig 3 and Fig 5.

When evaluating the manuscript, you stated that the analytical blank (46806 $^{14}C$ yr) seemed too low when compared to laboratory blanks. While it is possible to apply a much lower blank (e.g. 50,000 or 55,000) within the model, the model we use explicitly simulates foraminifera in the time domain, and thus requires a realistic $^{14}C$ activity to be assigned to all single foraminifera that are generated for all time periods. Since we don't know what the $^{14}C$ activity is beyond the end of *Marine13,* it is in practice not possible to use an analytical blank beyond the limit of *Marine13* without guessing/inferring what the Earth's $^{14}C$ activities were for periods beyond the limit of the calibration curve. Since we prefer not to do that, we use an analytical blank that is equivalent to the lowest activity in *Marine13.* This reasoning was absent from the original manuscript, so it has now been fully explained in the method section of the new manuscript, which will greatly assist the reader. The analytical blank is only relevant when interpreting Fig. 4, so we have provided additional information in the text when discussing Fig. 4 about how it can be interpreted in the case of a laboratory blank of e.g. 50,000 or 55,000 (the same principles apply).

We have also carried out other, minor changes requested by the referees, such as re-running the simulations with a measurement error of 500 $^{14}C$ yr (instead of 200) for old samples close to the analytical blank.

A "track changes" version of the manuscript is also appended for your convenience. As this track changes file was auto-generated by the computer by comparing two docx files, it might overstate the level of changes somewhat. Apologies for that.

Thank you again for your interest in our manuscript.

Kind regards,

Bryan Lougheed

[revised manuscript text omitted]